# An inter-comparison of inverse models for estimating European CH<sub>4</sub> emissions

Eleftherios Ioannidis<sup>1,\*</sup>, Antoon Meesters<sup>1</sup>, Michael Steiner<sup>2</sup>, Dominik Brunner<sup>2</sup>, Friedemann Reum<sup>3</sup>, Isabelle Pison<sup>4</sup>, Antoine Berchet<sup>4</sup>, Rona Thompson<sup>5</sup>, Espen Sollum<sup>5</sup>, Frank-Thomas Koch<sup>6,7</sup>, Christoph Gerbig<sup>7</sup>, Fenjuan Wang<sup>8</sup>, Shamil Maksyutov<sup>8</sup>, Aki Tsuruta<sup>9</sup>, Maria Tenkanen<sup>9</sup>, Tuula Aalto<sup>9</sup>, Guillaume Monteil<sup>10,+</sup>, Hong Lin<sup>11,12</sup>, Ge Ren<sup>11,12</sup>, Marko Scholze<sup>10</sup>, and Sander Houweling<sup>1,13</sup>

<sup>1</sup>Department of Earth Sciences, Vrije Universiteit, Amsterdam, The Netherlands

<sup>2</sup>Empa, Swiss Federal Laboratories for Materials Science and Technology, Dübendorf, Switzerland

<sup>3</sup>Deutsches Zentrum für Luft- und Raumfahrt e.V., Institut für Physik der Atmosphäre, Oberpfaffenhofen, Germany

<sup>4</sup>Laboratoire des Sciences du Climat et de l'Environnement, CEA-CNRS-UVSQ, Gif-sur-Yvette, France <sup>5</sup>NILU, Kjeller, Norway

<sup>6</sup>Meteorological Observatory Hohenpeissenberg, Deutscher Wetterdienst, Germany

<sup>7</sup>Department of Biogeochemical Signals, Max Planck Institute for Biogeochemistry, Jena, Germany

- <sup>8</sup>National Institute for Environmental Studies, Japan
- <sup>9</sup>Climate System Research, Finnish Meteorological Institute, Helsinki, Finland

<sup>10</sup>Department of Physical Geography and Ecosystem Science, Lund University, Lund, Sweden

<sup>11</sup>Division of Thermophysics Metrology, National Institute of Metrology, Beijing, China

<sup>12</sup>Zhengzhou Institute of Metrology, Zhengzhou, China

<sup>13</sup>SRON, Netherlands Institute for Space Research, Leiden, Netherlands

\*Now at: Research Development Satellite Observations, Royal Netherlands Meteorological Institute (KNMI), De Bilt, the Netherlands

\*Now at: Barcelona Supercomputing Center, Barcelona, Spain

Correspondence: Eleftherios Ioannidis (lefteris.ioannidis@knmi.nl, elefth.ioannidis@gmail.com)

## Abstract.

Atmospheric inversions are widely used to evaluate and improve inventories of methane  $(CH_4)$  emissions on scales ranging from global to national and beyond, combining observations with atmospheric transport models. This study uses the dense network of in situ stations of the Integrated Carbon Observation System (ICOS) to explore how well in situ data can constrain

5 European  $CH_4$  emissions. Following the concept of inter-comparison studies of the atmospheric tracer transport model intercomparison Project (TransCom), a  $CH_4$  inverse inter-comparison modeling study has been performed, focusing on Europe for the period 2006–2018. The aim is to investigate the capability of inverse models to deliver consistent flux estimates at the national scale and evaluate trends in emission inventories.

Study participants were asked to perform inverse modelling computations using a common database of a priori CH<sub>4</sub> emis-

10 sions and in-situ observations as specified in a protocol. The participants submitted their best estimates of  $CH_4$  emissions for the 27 European Union (EU) member states, the United Kingdom (UK), Switzerland, and Norway. Results were collected from 9 different inverse modelling systems, using 7 different global and regional transport models. The range of outcomes allows us to assess posterior emission uncertainty, accounting for transport model uncertainty and inversion design decisions, including a priori emission and model-data mismatch uncertainty.

25

- This paper presents inversion results covering 15 years, that are used to investigate the seasonality and trends of  $CH_4$  emissions. The different inversion systems show a range of a posteriori emission adjustments, pointing to factors that should receive further attention in the design of inversions such as optimising background concentrations. Most inverse models increase the seasonal cycle amplitude, by up to 400 Gg month<sup>-1</sup>, with the largest adjustments to the a priori emissions in Western and Eastern Europe. This might be due to underestimation of emissions from wetlands during summer or the importance of seasonality
- 20 in other microbial sources, such as landfills and waste water treatment plants. In Northern Europe, absolute flux adjustments are comparatively small, which could imply that the emission magnitude is relatively well captured by the a priori, though the lower station density could contribute also.

Across Europe, the inverse models yield a similar decreasing trend in  $CH_4$  emissions compared to the a priori emissions (-12.3% instead of -9.1%) from 2006 to 2018. While both the a priori and the a posteriori trend for the EU-27 are statistically significant from zero, their difference is not. On subregion scale, the differences between a posteriori and a priori trends are more statistically significant over regions with more in-situ measurement sites, such as over Western and Southern Europe.

Uncertainties in the a priori anthropogenic emissions, such as in the agriculture sector (cows, manure), or waste sector (microbial  $CH_4$  emissions), but also in the a priori natural emissions, e.g. wetlands, might be responsible for the discrepancies between the a priori and a posteriori emission trends in Western and Southern Europe. Our results highlight the importance of improving details in the inversion setup, such as the treatment of lateral boundary conditions and the model representation of

30 improving details in the inversion setup, such as the treatment of lateral boundary conditions and the model representation of measurement sites, to narrow the uncertainty ranges further.

#### 1 Introduction

Methane (CH<sub>4</sub>) is the second-most important anthropogenic greenhouse gas (GHG), after carbon dioxide (CO<sub>2</sub>), and has a significant contribution to global warming and climate change (*IPCC*, 2021). In the last two decades, CH<sub>4</sub> emissions increased by 20%, with concentrations reaching 1.923 parts per billion (ppb) in 2023 (*European Environment Agency*, 2022; World *Meteorological Organization (WMO)*, 2024). Globally, anthropogenic CH<sub>4</sub> emissions constitute 375 Tg yr<sup>-1</sup> or 50-60% of the total CH<sub>4</sub> emissions (*Saunois et al.*, 2024). The largest anthropogenic CH<sub>4</sub> emissions originate from agriculture (e.g., livestock production, rice cultivation), followed by the energy sector (fossil fuel production and use) and waste disposal (*IPCC*,

- 2021). However,  $CH_4$  is also emitted from various natural sources (248 Tg yr<sup>-1</sup>, Saunois et al. (2024)), with natural wetlands contributing up to 40% of the total  $CH_4$  emissions (Yusuf et al., 2012; Zhang et al., 2024). According to a comprehensive recent assessment, annual global  $CH_4$  emissions are around 575 Tg yr<sup>-1</sup> (Saunois et al., 2024). The Paris Agreement commits countries to implement mitigation measures to reduce GHG emissions. In addition, 150 countries have signed the Global Methane Pledge, launched in November 2021 at the Conference of the Parties (COP 26) with the aim of reducing global  $CH_4$
- emissions by 30% in 2030 relative to 2020 levels (Global Methane Pledge, 2023.).

Anthropogenic emission reporting is based on "bottom-up" inventories, and there are several bottom-up process-based models to estimate natural emissions and sinks. However, these anthropogenic and natural CH<sub>4</sub> emissions have large uncertainties (*Brandt et al., 2014; Zavala-Araiza et al., 2015; Deng et al., 2022; Arora et al., 2023)*. Uncertainties in anthropogenic emissions are caused primarily by uncertain emission factors used in bottom-up inventories (*Cheewaphongphan et al., 2019; Solazzo* 

- *et al.*, 2021). Some sources of anthropogenic emissions, such as fossil fuel, might also be missing from bottom-up inventories, as shown in a recent study by *Yu et al.* (2023). Process-based models of natural  $CH_4$  sources and sinks are uncertain for many reasons, including uncertain sensitivities to climatological conditions, small-scale variability that is difficult to scale up, and important processes that may still be missing (*Aalto et al.*, 2024). It is critical for countries to accurately quantify  $CH_4$  emissions, as there is a growing demand from policy makers, reinforced by the Paris Agreement, for efficient methods to reduce
- CH<sub>4</sub> emissions. Therefore, in addition to these bottom-up emission inventories and process-based models, "top-down" methods have been developed using inverse modeling techniques (*Bergamaschi et al., 2018a; Steiner et al., 2024*) to bring emission inventories into agreement with atmospheric measurements. These measurements provide independent information on emissions that can be used to evaluate emission inventories, through the use of inverse modeling, in support of the transparency framework of the Paris agreement (*World Meteorological Organization, 2016; Calvo Buendia et al., 2019*).
- The top-down approach, using inversion techniques, yields an optimised "a posteriori" estimate of the emissions. This is done by relating observed atmospheric dry air mole fractions to emissions using an atmospheric transport model, and by minimizing a Bayesian cost function with an inversion algorithm, starting from a priori information on emissions and their uncertainties (*Jacob*, 2007). Different techniques have been developed to solve the inverse problem, such as the Kalman smoother (*Bruhwiler et al.*, 2005), the ensemble Kalman filter (EnKF) (*Peters et al.*, 2005), and the 4D variational inversion (*Chevallier et al.*, 2005).
- Both EnKF and variational methods have advantages and disadvantages and are widely used today (*e.g. Bergamaschi et al.*, 2022; Saunois et al., 2019; Steiner et al., 2024).

Previous studies used the inverse modeling technique to estimate European CH<sub>4</sub> emissions, using regional (*Bergamaschi et al.*, 2018a, 2022; *Petrescu et al.*, 2023, 2024) or global (*Wang et al.*, 2019; *Deng et al.*, 2022; *Petrescu et al.*, 2023) transport models, based on in situ (e.g. *Bergamaschi et al.* (2022) or *Steiner et al.* (2024)) and satellite observations (e.g. *Bergamaschi* 

- *et al. (2013), Wang et al. (2019)). Bergamaschi et al. (2018b)* used different inverse models to estimate European CH<sub>4</sub> emissions for a period of six years (2006-2012). They showed a strong seasonality of CH<sub>4</sub> emissions in Europe due to wetland emissions. In a more recent study, *Bergamaschi et al. (2022)* focused on 2018 using three high resolution inverse models that showed a posteriori emissions were higher in Germany and the Benelux than the emissions reported to the United Nations Framework Convention on Climate Change (UNFCCC).
- Here, we present a new inverse modelling inter-comparison study, with the aim of estimating European  $CH_4$  emissions over the period 2005-2019. We used a combination of in situ measurement databases, most importantly from the extended Integrated Carbon Observation System (ICOS) network. The major objective is to evaluate and compare the performance of the nine inverse models participating in the inter-comparison. This study uses the extended measurement time series to estimate trends in total  $CH_4$  emissions in Europe until 2019. In addition, we try to address the systematic difference in emission seasonality re-
- ported by Bergamaschi et al. (2018b). Previous studies have shown large discrepancies between inversion-estimated emissions

of CH<sub>4</sub> (*Petrescu et al.*, 2021, 2023). To better understand these differences and to eliminate some of the potential causes, our experimental protocol (*Florentie and Houweling*, 2021), presented in Sect. 2, prescribes the a priori emissions and observations to be used. The a priori emissions, the observations used for the different simulation experiments, the validation dataset, the participating models, and the simulations carried out are described in Section 3. Information about the modelled output databases is also provided in Sect. 3. The results and a discussion of our findings are presented in Sect. 4. The implications of our findings are presented in Conclusions (Sect. 5).

#### 2 Inversion Protocol

To assess European CH<sub>4</sub> emissions using an ensemble of inversions, a protocol has been formulated by *Florentie and Houweling (2021)*, which the participants are required to use. It closely follows a protocol established in the VERIFY project 90 (https://verify.lsce.ipsl.fr/) and utilizes datasets that were collected as part of it. The participants have been instructed to use 91 only atmospheric observations from common datasets (see Sect. 3.1) and a common set of a priori CH<sub>4</sub> emissions (see Sect. 3.2). The protocol also provides climatological radon (<sup>222</sup>Rn) fluxes (*Karstens et al., 2015*) for simulating radon, to assess 92 the performance of the atmospheric transport models that are used. The groups running regional models are required to use 93 initial and lateral boundary conditions from the Copernicus Atmosphere Monitoring Service (CAMS) CH<sub>4</sub> reanalysis v19r1

- (*Agustí-Panareda et al., 2023*), based on assimilated surface observations. Two inversion systems use the Rodenbeck 2-step inversion approach (*Rödenbeck et al., 2009*), for which consistent baseline conditions are made available as part of the protocol. However, the protocol does not specify the meteorological boundary conditions, or the background, observation, and a priori emissions uncertainties to be used, and whether or not to optimise background concentrations. The participants are requested to provide monthly gridded  $CH_4$  fluxes at 25 km<sup>2</sup> grid spacing, a priori and a posteriori national total emissions, mole fraction
- time series at the measurement sites and their uncertainties. National total emissions are to be provided for at least the European Union (EU-27) countries, the United Kingdom (UK), Norway, and Switzerland. Regional inversions should cover at least the area from  $15^{\circ}$ W to  $35^{\circ}$ E and  $35^{\circ}$ N to  $70^{\circ}$ N. The inversions should cover as many years as possible from 2005 to 2019. In case it is not possible to provide results for the full period, then the groups are asked to submit results for a selection of years, chosen to cover the full period as well as possible, including at least the years 2008, 2013 and 2018. This study focuses on total CH<sub>4</sub>
- emissions, i.e. without sectorial separation of the a posteriori fluxes.

#### 3 Methodology

#### 3.1 Atmospheric measurements

The European monitoring stations used in this study are shown in Figure 1 and additional information is provided in Table A1 in Appendix A. The observations are made available by the Integrated non- $CO_2$  Greenhouse gas Observing System (InGOS)

project (2005-2018) (*INGOS, 2018*), the National Oceanic and Atmospheric Administration (NOAA) flask sampling network in Europe (2005-2018) (*Lan et al., 2023*), the Advanced Global Atmospheric Gases Experiment (AGAGE), the ICOS network