# Peer review of "An inter-comparison of inverse models for estimating European CH4 emissions"

_Earth System Science Data, 2025_

## Author Comment (AC1)

**Responses to Reviewers**

We thank both Reviewers for their comments. First we provide the reviewers' text and below our reply in blue. We first provide our replies to Reviewer 1, followed by our replies to Reviewer 2. At the end of the document, we list the new references we included in the revised manuscript. Other minor corrections are included in the revised manuscript.

**Response to Reviewer 1**

This study presents an interesting comparison of nine inverse models for estimating national CH4 emissions across Europe from 2005 to 2018. The topic is cutting-edge and highly relevant for evaluating the effectiveness of nationwide greenhouse gas reduction efforts, as well as assessing the accuracy of bottom-up emission inventories. The paper is well-structured and scientifically sound in most parts; however, several key concerns should be addressed before considering acceptance in ESSD. I recommend a **major revision** based on the following specific comments:

**Data resolution and coverage**:

The prior CH4 emissions used in this study vary in resolution and temporal coverage (Table 1). For instance, the Peatlands category has a daily resolution, while other sources are provided at monthly or annual resolutions. The authors should justify their selection of these datasets and explain how missing time periods (e.g., 2005–2020 vs. 2005–2018) were handled.

As discussed in the protocol (https://doi.org/10.5281/zenodo.15082281, 2021) it was decided to use the same dataset as in the VERIFY project to be able to compare the current output fluxes with results from previous projects. Natural emissions from JSBACH-HIMMELI were prepared as part of the CoCO$_2$ project. For anthropogenic emissions, the year 2018 is used for years 2019 and 2020. We modified the text as follows:

"A priori CH$_4$ emissions used in this study are summarised in Table 1 including information on their spatial and temporal resolutions. The same bottom-up inventories and process-based models for generating natural emissions are used here as in the EU H2020 VERIFY project. More specifically, for anthropogenic CH$_4$ emissions, the Emissions Database for Global Atmospheric Research (EDGAR) v6.0 is used, which provides emissions for different anthropogenic sectors (Monforti et al., 2021). Year 2018 is repeated for 2019 and 2020. For the anthropogenic emissions, the protocol does not provide information on monthly, daily, or hourly factors to scale the emissions, so all models used temporally constant values. Natural CH$_4$ emissions from peatlands and mineral soils derived from the JSBACH-HIMMELI model (Petrescu et al., 2023), prepared as part of the EU H2020 CoCO$_2$ project and do account for seasonality. Climatological CH$_4$ emissions from inland water, termites, ocean, and geological sinks/sources are used as shown in Table 1. Monthly climatological emissions from ocean and inland water are repeated for all years. The ULB emissions for inland water are provided by the VERIFY project. " (Lines: 127 - 136)

**Model validation:**

For the validation of prior and posterior CH4 estimates against observations, only six out of nine inverse models are presented in Figure 5. Why were the remaining three models excluded?

Thank you for your comment. We did not exclude any model in Fig. 5 (now Fig. 6). Not all the modeling groups provided validation data, as shown in Table 4. We added a sentence to remind the reader in the manuscript:

"Bear in mind that not all inverse models provided results for the validation stations (see the 'Validation data' part in Table 4)." (Lines: 243 - 244)

**CTDAS-WRF model performance (Line 230):**

The authors attribute the poorer performance of the CTDAS-WRF model to discrepancies with observations during winter and fall. Does this discrepancy also apply to other inverse models? Beyond meteorological variability, what other factors (e.g., transport errors) might contribute?

Thank you for your comment. Of course this discrepancy could apply to other inverse models. However, the performance of the other inverse models is decent and the modellers did not report any model discrepancies with the observations, similarly to CTDAS-WRF.
We state in Lines 236-237 in the submitted manuscript that errors in simulating shallow boundary layer is a common transport model error. We added the following text:

"Errors in the modeling of atmospheric transport, such as advection schemes, sub-grid scale parameterizations, and limited horizontal and vertical resolutions, could also be responsible for these discrepancies, as has been reported by previous studies, such as Locatelli et al. (2013)." (Lines: 253 - 255)

**Seasonal cycle (Figure 6):**

One inverse model exhibits abnormal seasonality compared to others, particularly in Western Europe (August–December). What explains this large variability? This likely offsets the posterior seasonality, especially in August. A more detailed explanation is needed here.

In Southern Europe, the sharp decline in mean posterior emissions from August to November appears driven by one outlier model, potentially biasing the seasonality interpretation.

Two inverse models exhibit abnormal seasonality, CTDAS-WRF (Western Europe) and NTLB (Southern Europe). It is difficult to point out the reason for this behaviour in the different sub-regions, without detailed sensitivity simulations. We can only speculate. As has been mentioned in the text CTDAS-WRF exhibits the worst performance when validated against independent data (old Fig. 5 - now Fig. 6) and we assume this is due to uncertainties on transport (PBL dynamics). NTLB also uses WRF. It is known that WRF has difficulties simulating realistic PBL mixing and structure and strongly depends on the parametrisation used (see for example https://doi.org/10.1016/j.scitotenv.2016.07.167 or DOI: 10.20937/ATM.2016.29.01.05). These discrepancies shown by CTDAS-WRF and NTLB

could be due to difficulties of WRF simulating transport realistically at 0.25 x 0.25 degrees spatial resolution. We don't think it is appropriate to remove inverse models from the comparison for this reason. The data are publicly available and the reader is free to use the results they want for their purposes. However, we discuss in the text about the outlying models and the reasons we think are responsible for this behaviour. We also provide a discussion on the seasonality if we remove these two models. We provide the figure without CTDAS-WRF and NTLB at the end of this document. We add the following text to address reviewers' concerns:

"Two inverse models exhibit abnormal seasonality: CTDAS-WRF in Western Europe (August-December) and NTLB in Southern Europe (from August to November), despite the latter performing better than CTDAS-WRF when comparing against the independent observations (Fig. 6). CTDAS-WRF and NTLB are driven by the same transport model (WRF), although both inverse models use a different inversion setup as shown in Table 3. This makes it difficult to point out the cause of these discrepancies. It is known from the literature that WRF has difficulties simulating realistic PBL mixing and structure and its performance varies with the PBL scheme that is used (e.g. Banks and Baldasano (2016)). We estimated the seasonal cycle without those two inverse models (not shown here). Then the results show a stronger peak during spring and summer in EU-27. When excluding results from CTDAS-WRF and NTLB, the seasonal patterns remain largely consistent across Southern, Northern, Western and Eastern Europe, with minimal changes to the overall seasonality. " (Lines: 332 - 340)

Regarding Northern Europe (Lines 255–260), the authors attribute enhanced prior CH4 emissions to wetlands in summer. Could seasonal variations in $CH_4$ sinks such as ●OH also play a role?

Thank you for your comment. We added the following text:

"We expect the influence of the hydroxyl radical (·OH) on $CH_4$ to be small over Europe (Zhao et al., 2020). East et al. (2024) attributed wetland emissions as the primary driver of $CH_4$ seasonality during summer in the northern hemisphere, while $CH_4$ sinks, such as ·OH , are unlikely to play a significant role."  (Lines: 284 - 287)

Here is the paper by East et al. 2024: https://doi.org/10.1029/2024GL108494

For the JSBACH-HIMMELI model, the authors acknowledge underestimation of river and lake emissions. Are coastal wetland emissions well-captured in this model?

Thank you for your comment. JSBACH-HIMMELI does not explicitly resolve coastal wetland emissions. Coastal wetland emissions appear in JSBACH-HIMMELI emission maps as part of 'wet mineral land' or 'inundated land' and the emissions are calculated using the approach by Spahni et al. (2012) and the emission results have not been compared against measurements in coastal wetlands.

We added the following text in the manuscript:

"Though JSBACH-HIMMELI does not explicitly resolve coastal wetland emissions." (Lines: 290-291)

Lines 260–265: The authors note that uncertainties in temperature and precipitation limit wetland $CH_4$ emission estimates. However, precipitation is a poor proxy for wetland emissions compared to inundation (see https://doi.org/10.1029/2020GB006890 and https://doi.org/10.1038/s43247-025-02438-3). A discussion of these hydrological indicators would strengthen the analysis.

Thank you for your comment and sharing these two interesting papers which show inundation to be important for tropical wetlands. We agree that inundation could result in high $CH_4$ emissions during spring in northern Europe. We added the following text:

"Although temperature and precipitation are important drivers, studies suggest that $CH_4$ emissions are more sensitive to inundation (Gerlein-Safdi et al., 2021). Inundation, after snow-melt, could induce large $CH_4$ emissions in spring. Inundation in JSBACH-HIMMELI is taken as prescribed from satellite data (WAD2M, Zhang et al. 2021) and $CH_4$ emissions from inundated lands are calculated using the approach by Spahni et al. (2011). However, bottom-up process-based models, such as JSBACH-HIMMELI, have limitations combining emissions from different types of land, which might result in limitations in the total wetland $CH_4$ emissions." (Lines: 294 - 299)

Suggestion: Replace "missing processes" (Line 264) with "missing/simplified processes" to account for parametrization simplifications in bottom-up models.

Thank you, we did.

Lines 279–281: The deduction that wetland emissions may decrease in Southern/Eastern Europe due to reduced precipitation is problematic. Precipitation is a weak predictor of wetland CH4 emissions due to time lags (runoff, microbial decomposition). Inundation or GRACE terrestrial water storage data would be more appropriate. The argument for future projections based on precipitation should be reconsidered.

Thank you for your comment. Regarding the model: Inundation in JSBACH-HIMMELI is taken as prescribed (WAD2M satellite-data - based product, Zhang et al 2021) and $CH_4$ emissions from inundated lands are calculated using the approach by Spahni et al (2011), using JSBACH respiration fluxes in the calculation.

Precipitation and inundation are of course linked. Precipitation may seem like a poor proxy because of this delay and dynamics related to snow melt, but it is nevertheless an environmental driver. Also inundation, as the inundation extent depends on snow melt/precipitation during previous days and weeks. Inundation extent directly scales the methane emission but it is not the only process which gives rise to methane emissions, as peatlands and mineral soils which are not in inundation can also emit methane. The models generally, and also JSBACH, have limitations in combining emissions from different types of land.

However, we decided to remove our statement about precipitation in the paper to avoid any confusion and getting into a detailed discussion which will be out of the scope of this paper.

Lines 296–299: The seasonal variability in Western Europe is linked to agricultural and fossil fuel emissions. Is the agricultural sector large enough to drive summer seasonality? Source-resolved posterior analyses could clarify this.

This is only an assumption from our side. Agriculture emissions, such as livestock and manure management, are a dominant sector in Western Europe. Emissions of storage and treatment of manure are temperature dependent, and exhibit seasonal variations that might not be well accounted for in the bottom-up inventories. Unfortunately, a source-resolved posterior analysis using all the inverse models is not possible, as these detailed posterior fluxes are not provided as an output from all inverse models/systems. In addition, methane measurements provide limited information to constrain the emission from specific sectors. This could further be investigated in upcoming inter-comparison projects.

We added the following text in the revised manuscript:

"Conducting source-resolved posterior analyses in future studies, for example using isotopic measurements, would facilitate a more precise quantification of contributions from agricultural and fossil fuel sources." (Lines: 329- 331)

**Interannual trends (Figure 7):**

Why do negative emissions appear? Are these CH4 emission anomalies? Clarify how anomalies were calculated. The figure caption and y-axis title need revision for accuracy.

Thank you for your comment and our apologies for missing out on this. Indeed we show emission anomalies. The same is the case also for the seasonality. We have updated the figures, figures caption and we added the following text for the trends:

"Standarised anomalies are first estimated by inverse model and then the results are averaged to get the mean posterior anomaly trends." (Lines: 357 - 358)

**Underestimated wetland emissions (Line 370):**

A quantitative estimate (e.g., "underestimated by ~20%") would strengthen the discussion.

Thank you we added a percentage as the Reviewer suggested.

**Background concentration influence (Line 386):**

The impact of background concentration determination on posterior estimates is underdiscussed and should be expanded.

We added the following text:

"The choice of optimising background concentrations seems to be important for constraining long-range transport or inconsistencies caused by the lateral boundary conditions. Including background concentration optimization within the inversion framework enhances the agreement between posterior modelled and observed concentrations, particularly in regions close to the boundaries of the modeling domain and minimises uncertainties due to biases on long-way transport (Steiner et al., 2024)." (Lines: 433 - 437)

**Study implications:**

The broader implications of this work—particularly strategies to reduce inter-model discrepancies—should be discussed.

We added the following text:

"Detailed protocols with prescribed prior emissions, common observations to be used for optimisation and validation, and lateral boundary conditions, as has been done in this study, can help to narrow down inter-model discrepancies. The use of common meteorological boundary conditions in a subset of inverse models, as in Munassar et al. (2023), could be explored to shed light on the causes of transport errors in the models." (Lines: 441 - 444)

**Minor Corrections:**

Figure 3 caption: Inconsistent with subplot order.

Thank you, we corrected the order.

Line 328: "EU 27, where they a show similar strong negative trend" correct the typo to "EU-27, where they show a similarly strong negative trend."

Thank you. We corrected the typo and changed "EU27" to "EU-27" throughout the manuscript.

**Response to Reviewer 2**

Ioannidis et al. performed a CH4 inverse inter-comparison modeling (MIP) study to estimate European CH4 emissions. With a suite of inverse models that use different transport models and have different model resolutions, designs of state vectors, and data assimilation techniques, they investigate differences in their estimated emission magnitudes, spatial patterns, seasonal variations, and trends. Given that atmosphere-based estimates are considered as an important tool to assess the accuracy of national greenhouse gas inventory reporting and their external uncertainties are often hard to quantify by using one modeling system alone, such an inter-comparison modeling study provides important insights on how well the atmospheric observations can be used to quantify European CH4 emissions. The current manuscript is well organized. It well describes the MIP protocols, the participating modeling systems, and their obtained results. It can be further improved by some revision.

1. The authors discussed similarities and differences in the adjustments of the posterior estimates relative to the common prior based on different inversion results (e.g. Fig 2). Although such information is useful, it is also important to know, with the posterior adjustments, whether posterior emissions show similar spatial patterns as the prior. Therefore, besides Fig. 2, maps that show posterior emissions, as well as some discussion on the posterior spatial patterns, could be useful to add.

   Thank you for your comment. We now show the posterior fluxes from all the inverse models in the Appendix (Appendix C) and we added the following text in the main manuscript:

   "Figure C1 in Appendix C shows the a posteriori $CH_4$ fluxes for all the inverse models. The spatial distribution of posterior $CH_4$ fluxes are similar for all the inverse models and the prior fluxes over the Benelux region, south of Poland, Finland, the UK and Bretagne. Similar spatial patterns are shown between the a priori fluxes and for all inverse models except for CIF-FLEXPART over Romania and Po Valley and in the North Sea except for CSR. However, the posterior emission adjustments compared to the prior fluxes are shown much clearer by calculating their differences." (Lines: 165 - 169)

   "In the TEST simulation all the a posteriori fluxes show spatial patterns that are similar to the a priori fluxes, such as over the Benelux region, Po Valley, Romania and southern Poland. " (Lines: 206 - 207)

2. The authors evaluated the performance of inverse models with observation used in the optimization and independent data. It is obvious that the performance varies quite a bit with the ICONDA model standing out as the best in near all the statistics. This is useful to know. However, it would be more beneficial for the community if the authors can provide specific insights on why ICONDA performs the best. Is it due to more accurate transport simulations, or their data assimilation techniques, or the optimization of their boundary values, or the suitability of their specified error covariance parameters? Although the authors mentioned about the importance of atmospheric transport modeling, they rarely mention the importance of the error

covariance parameters at all. To me, the relatively poorer performance of the CTDAS-WRF may relate to the possibility of an overfitting of their observations.

Thank you for your comment. We understand the reviewer's comment about ICONDA's performance, however we decided only to refer to the recent paper which describes the model's development and detailed tests for readers to go through if they are interested because we believe it's out of the scope of this paper to state the reasons inverse models (ICONDA, LUMIA) perform the best. We now have expanded the discussion only about the poorer performance of CTDAS-WRF which is the less tested inverse model for $CH_4$.

About CTDAS-WRF:

"Two of the inverse models that submitted results for the validation stations simulated these observations considerably less well than the observations that were optimised: CSR (Fig. 5c) and CTDAS-WRF (Fig. 5a,c), where the latter shows the poorest performance in this metric among all models, despite using a similar inversion setup to an outperforming inverse model, such as ICONDA (see Table 3). The poorer overall performance of CTDAS-WRF is driven by big discrepancies with the observations during winter and fall (not shown here). Hence, we assume that this could be due to errors in simulating the shallow boundary layer, which is a common transport model error (Gerbig et al., 2008; Deng et al., 2017; Lehner and Rotach, 2018). Errors in the modeling of atmospheric transport, such as advection schemes, sub-grid scale parameterizations, and horizontal and vertical resolutions, could also be responsible for these discrepancies, as has been reported by previous studies, such as Locatelli et al. (2013). Complex terrain, e.g. mountainous sites, could also introduce biases in the results, as it is difficult to simulate inflow in and around mountains (e.g. Oqaily et al . 2025). The performance of CTDAS-WRF with respect to the stations that were optimized for is much better (compare Figure 5 and Figure 6), which suggests that the poor performance with respect to the validation stations could be due to overfitting. However, on average, the fit to the optimized stations does not improve more in the CTDAS-WRF inversion than some other models (Figure 5), suggesting that the weight of the observations in the inversion was not considerably larger than in the other models. Overfitting could still play a role for individual stations, but further analysis is needed. Finally, the statistics presented here are averaged using all the stations for the common years. Therefore it is possible that the statistics are driven by one of the stations." (Lines: 247 - 262)

3. Base versus test runs. It is nice to see the authors conducted both base versus test runs to assess whether the posterior emission estimates can be improved with additional sites. However, it is very hard to compare the performance of the base versus test runs in their current presentation. Please consider to add the summary statistics for both base and test runs into the same barcharts. For example, merge Figs. 4 and D1 and merge Figs 5 and E1. Also, the authors only mentioned the summary statistics in those figures without much discussion. Please add discussion on the test runs.

Thank you for your comment. We included two tables in section 4.2, which summarises the statistics for the BASE and TEST runs and only focuses on the 6

inverse models that provided results for both simulations. We removed Figs. D1 and E1. We also added the following discussion on the TEST runs:

For the optimised stations:

"Table 5 summarises the statistics for the inverse models that provide $CH_4$ mole fractions for the optimised stations in the BASE and TEST simulations. The use of more stations results in improved statistics for all inverse models in general. For example, RMSE is further improved for all inverse models in the TEST simulation, with comparable correlation coefficients between the two runs. ICONDA and LUMIA performed better than the other two inverse models. " (Lines: 236 - 240)

For the validation stations:

"Table 6 summarises the statistics for the inverse models that provide $CH_4$ mole fractions for the validation stations for both the BASE and TEST simulations. The use of more stations in the TEST simulation resulted in better agreement between the modelled and the observed molar fractions for all inverse models, as shown by the lower RMSEs for all models. ICONDA performs better than the other two inverse models with a lower bias and a higher correlation coefficient in both simulations. For details regarding ICONDA's development and detailed testing for European CH4 inversions please refer to Steiner et al. (2024). " (Lines: 262 - 267)

4.  For the results discussing the seasonal cycle and trend. Consider add additional lines summarizing the posterior results averaged among the best performing models (e.g. ICONDA, NTLB, and another one?). It would be interesting to know if they will get the same seasonal cycle or trend by only using the best performing models. Also, consider to add some discussion on these results too.

Thank you for your comment. There are two inverse models that exhibit abnormal seasonality, as pointed out by Reviewer 1. These inverse models are: CTDAS-WRF in Western Europe and NTLB in Southern Europe. Therefore we have added the following text summarising the averaged posterior results by removing these two inverse models. Bear in mind this is applicable only for the seasonal analysis and not for the trends as these two inverse models, as well as ICONDA, are not included in the trends analysis because they haven't submitted results for the full period as shown in Table 4 and mentioned in Lines 313-315 in the submitted manuscript.

"Two inverse models exhibit abnormal seasonality: CTDAS-WRF in Western Europe (August-December) and NTLB in Southern Europe (from August to November), despite the latter performing better than CTDAS-WRF when comparing against the independent observations (Fig. 6). CTDAS-WRF and NTLB are driven by the same transport model (WRF), although both inverse models use a different inversion setup as shown in Table 3. This makes it difficult to point out the cause of these discrepancies. It is known from the literature that WRF has difficulties simulating realistic PBL mixing and structure and its performance varies with the PBL scheme that is used (e.g. Banks and Baldasano (2016)). We estimated the seasonal cycle without those two inverse models (not shown here). Then the results show a stronger peak during spring and summer in EU-27. When excluding results from CTDAS-WRF and NTLB, the seasonal patterns remain largely consistent across Southern,

Northern, Western and Eastern Europe, with minimal changes to the overall seasonality." (Lines: 332 - 340)

5.  In the trend section, it would be insightful to know what drives the declining trend in CH4 emissions over eastern and southern Europe.

    Thank you for your comment. It is a bit difficult to know what drives the stronger declining trend over Eastern and Southern Europe from this analysis. These are the results we are getting when using in-situ data. We assume the in-situ stations provide different information compared to what the a priori emission inventories report and therefore we get the stronger trends.  We added the following text in the manuscript:

    "The stronger decline on $CH_4$ emissions over Southern and Eastern Europe compared to the a priori could be driven from the observations used to constrain the a priori emissions. Follow up studies could further explore what drives the emission trends (e.g. in-situ stations, background optimisation) in the European boundaries. " (Lines: 376 - 378)

Other:

Line 151: "mg m-2 hr-1" - this is the unit used for the maps, not for the trends and seasonal cycles.

Thank you, we removed the units from this line.

Fig. 1 – label the country names. This would be useful for readers to link the country-based similarities or differences to specific areas in the maps.

Thank you, we did.

Fig. 3 – please indicate where the additional sites are considered

Thank you, we did.

[Figure]

Similar to Fig.7 in the revised manuscript, but without CTDAS-WRF and NTLB inverse models. The posterior fluxes per inverse model are shown in dashed lines: ICONDA is shown in green, CTE-CH$_4$ in pink, NTFVAR in orange, CIF-CHIMERE in gray, LUMIA in brown, CSR in yellow and CIF-FLEXPART in red.

New references included in the revised manuscript:

Al Oqaily, D., Giani, P., & Crippa, P. (2025). Evaluating WRF multiscale wind simulations in complex terrain: Insights from the Perdigão field campaign. Journal of Geophysical Research: Atmospheres, 130, e2025JD044055. https://doi.org/10. 1029/2025JD044055

Banks, R. and Baldasano, J.: Impact of WRF model PBL schemes on air quality simulations over Catalonia, Spain, Science of The Total Environment, 572, 98–113, https://doi.org/https://doi.org/10.1016/j.scitotenv.2016.07.167, 2016

East, J. D., Jacob, D. J., Balasus, N., Bloom, A. A., Bruhwiler, L., Chen, Z., Kaplan, J. O., Mickley, L. J., Mooring, T. A., Penn, E., Poulter, B., Sulprizio, M. P., Worden, J. R., Yantosca, R. M., and Zhang, Z.: Interpreting the Seasonality of Atmospheric Methane, Geophysical Research Letters, 51, e2024GL108 494, https://doi.org/https://doi.org/10.1029/2024GL108494, 2024.

Gerlein-Safdi, C., Bloom, A. A., Plant, G., Kort, E. A., & Ruf, C. S. (2021). Improving representation of tropical wetland methane emissions with CYGNSS inundation maps. Global Biogeochemical Cycles, 35, e2020GB006890. https://doi. org/10.1029/2020GB006890

Locatelli, R., Bousquet, P., Chevallier, F., Fortems-Cheney, A., Szopa, S., Saunois, M., Agusti-Panareda, A., Bergmann, D., Bian, H., Cameron-Smith, P., Chipperfield, M. P., Gloor, E., Houweling, S., Kawa, S. R., Krol, M., Patra, P. K., Prinn, R. G., Rigby, M., Saito, R., and Wilson, C.: Impact of transport model errors on the global and regional methane emissions estimated by inverse modelling, Atmos. Chem. Phys., 13, 9917–9937, https://doi.org/10.5194/acp-13-9917-2013, 2013.

Munassar, S., Monteil, G., Scholze, M., Karstens, U., Rödenbeck, C., Koch, F.-T., Totsche, K. U., and Gerbig, C.: Why do inverse models disagree? A case study with two European $CO_2$ inversions, Atmos. Chem. Phys., 23, 2813–2828, https://doi.org/10.5194/acp-23-2813-2023, 2023.

Spahni, R., Wania, R., Neef, L., van Weele, M., Pison, I., Bousquet, P., Frankenberg, C., Foster, P. N., Joos, F., Prentice, I. C., and van Velthoven, P.: Constraining global methane emissions and uptake by ecosystems, Biogeosciences, 8, 1643–1665, https://doi.org/10.5194/bg-8-1643-2011, 2011.

Zhao, Y., Saunois, M., Bousquet, P., Lin, X., Berchet, A., Hegglin, M. I., Canadell, J. G., Jackson, R. B., Dlugokencky, E. J., Langenfelds, R. L., Ramonet, M., Worthy, D., and Zheng, B.: Influences of hydroxyl radicals (OH) on top-down estimates of the global and regional methane budgets, Atmos. Chem. Phys., 20, 9525–9546, https://doi.org/10.5194/acp-20-9525-2020, 2020.